# A Flow-Through Microfluidic Relative Permittivity Sensor

**DOI:** 10.3390/mi11030325

**Published:** 2020-03-20

**Authors:** Yaxiang Zeng, Remco Sanders, Remco Wiegerink, Joost Lötters

**Affiliations:** 1Integrated Devices and Systems group, University of Twente, P.O. Box 217, 7500 AE Enschede, The Netherlands; r.g.p.sanders@utwente.nl (R.S.); r.j.wiegerink@utwente.nl (R.W.); j.c.lotters@utwente.nl (J.L.); 2Bronkhorst High-Tech B.V., Nijverheidsstraat lA, 7261 AK Ruurlo, The Netherlands

**Keywords:** capacitance sensor, interdigitated electrodes, relative permittivity sensor

## Abstract

In this paper, we present the design, simulation, fabrication and characterization of a microfluidic relative permittivity sensor in which the fluid flows through an interdigitated electrode structure. Sensor fabrication is based on an silicon on insulator (SOI) wafer where the fluidic inlet and outlet are etched through the handle layer and the interdigitated electrodes are made in the device layer. An impedance analyzer was used to measure the impedance between the interdigitated electrodes for various non-conducting fluids with a relative permittivity ranging from 1 to 41. The sensor shows good linearity over this range of relative permittivity and can be integrated with other microfluidic sensors in a multiparameter chip.

## 1. Introduction

Microfluidic relative permittivity sensors have great potential for fluid discrimination and composition determination [1,2]. Applications include the detection of medicine composition in medical infusion pumps [3], single-cell impedance cytometry [4,5], composition detection in flow chemistry [6,7], production monitoring of polymers [8], void-fraction measurement in two-phase flow [9,10], and measurement of glucose concentration in water [11,12] or water content in oil [13,14].

To measure relative permittivity, an electric field or electromagnetic wave needs to be applied. Quite often, this is done by using a planar electrode structure, with two electrodes next to each other [2] or with interdigitated electrodes [1,13,15]. These designs can be relatively small in size. However, the electric field in these designs is usually limited to the side of the fluidic channel. A much larger capacitance with less parasitic capacitance can be realized by using parallel plate structures, where the fluid is flowing in between the electrodes [3,6,16], but this results in a trade-off between channel diameter and capacitance. Furthermore, most of the current designs require a significant amount of chip area as they use channels parallel to the chip surface. A comparison between previously reported devices and the device presented in this paper is given in Table 1.

In this paper, we present a device where the fluid is flowing in between interdigitated electrodes, allowing for both a large channel diameter and large capacitance. The sensor can, for instance, be bonded above a fluid inlet or outlet of an existing sensor, such as a micro Coriolis flow sensor [17]. As such, it is very suitable for combination with other sensors in the same package to form an integrated multi-parameter sensor system as proposed in [3].

## 2. Design and Simulation

Figure 1a shows a cross-sectional schematic drawing of the proposed device. The design is based on using an SOI wafer with a 400 μm-thick handle layer, a 2 μm thick buried oxide (BOX) layer, and a 25 μm thick device layer. Openings of 90 μm × 500 μm are etched through the handle layer and form the fluidic inlet and outlet. Each inlet or outlet consists of two parallel openings at a distance of 350 μm from each other, as shown in the top view in Figure 1b. The distance between the inlet and corresponding outlet openings is 3.5 mm. Each chip contains two complete devices. Interdigitated electrodes are suspended in the device layer above each inlet and outlet opening, so that each chip contains eight independent pairs of sensor electrodes. In Figure 1a and Figure 2, the electrodes are indicated by a red and green color. The fluid flows through the gaps between the electrodes and the capacitance between the electrodes is directly proportional to the relative permittivity of the fluid. A MEMpax glass cap with a fluidic channel is glued on top of the silicon chip and forms the fluidic connection between the inlet and outlet.

Figure 2 shows the mask layout for the interdigitated electrodes. The 90 μm × 500 μm inlet or outlet opening is indicated by the dashed outline. The electrodes are electrically connected from one side and at the other side mechanically anchored to the BOX layer by 50 μm × 50 μm anchor structures. Isosceles trapezoid shaped fingers are added to both sides of each electrode to further increase the capacitance. The short base, long base and the height of the isosceles trapezoids are 5 μm, 13.3 μm and 12.5 μm respectively. The center to center distance between the trapezoids is 20 μm. Outside the inlet or outlet area, the electrodes are electrically isolated by a 5 μm wide trench.

Figure 3 shows a lumped-element electrical circuit of the entire relative permittivity sensor. Table 2 provides an overview of the components in this circuit with their measured or calculated values.

To measure the impedance, the sensor is driven by an AC voltage $V_{i}$ at one side while the current $I_{o}$ is measured at the other side, as indicated in Figure 3. In that case, the influence of $C_{P1}$ and $C_{P4}$ can be neglected. The series resistors $R_{P1}$ and $R_{P2}$ are smaller than 48 Ω. For frequencies up to 1 MHz, this is much smaller than the impedance of $C_{P2}$, $C_{P3}$ and the fluid dependent capacitance $C_{1}$. Therefore, for measuring frequencies up to 1 MHz, $R_{P1}$, $R_{P2}$, $C_{P2}$ and $C_{P4}$ can also be neglected. The native oxide on the interdigitated electrodes is only a few nm thick so $C_{o1}$ and $C_{o2}$ are very large and their impedance can be ignored. Thus, for frequencies up to 1 MHz, it can be assumed that the measured impedance is given by:(1)$$\begin{array}{cc} Z & =\left(C_{1}+C_{P}\right)\|R_{1}\\ \; & =\frac{R_{1}}{-j\omega R_{1}\left(C_{1}+C_{P}\right)+1} \end{array}$$

With the reactance $X_{C_{1}+C_{p}}$ of $C_{1}$ and $C_{P}$ given by:(2)$$X_{C_{1}+C_{p}}=-\frac{1}{\omega (C_{1}+C_{P})}$$
we find the following expression for the real part $Z^{\prime }$ and imaginary part $Z^{\prime \prime }$ of the impedance Z:(3)$$Z^{\prime }=\frac{{R_{1}}^{2}}{{R_{1}}^{2}+{X_{C_{1}+C_{p}}}^{2}}\;\cdot \;\frac{{X_{C_{1}+C_{p}}}^{2}}{R_{1}}$$
(4)$$Z^{\prime \prime }=\frac{{R_{1}}^{2}}{{R_{1}}^{2}+{X_{C_{1}+C_{p}}}^{2}}\;\cdot \;X_{C_{1}+C_{p}}$$

The parasitic parallel capacitance $C_{P}$ is assumed constant and the capacitance $C_{1}$ will be proportional to the relative permittivity $\varepsilon_{r}$ of the fluid:(5)$$C_{1}=\varepsilon_{0}\varepsilon_{r}k$$
For a parallel plate capacitor the proportionality constant *k* would be equal to the ratio between the surface area of the electrodes and the distance between them, however, because of the complex geometry of the device, a parallel plate approximation cannot be used. Therefore, finite element simulations were done with COMSOL Multiphysics® to estimate the capacitance and the constant *k* in Equation (Equation 5). The electrodes consist of 13 identical elements, so only one element as shown in Figure 4a needs to be used in the simulations. In air, the interdigitated electrode capacitance above each of the inlet or outlet openings was estimated to be 0.187 pF, corresponding to a value of *k* of 21 mm. Figure 4b shows the simulated electric potential between electrodes.

## 3. Sensor Fabrication

Figure 5 shows the fabrication process of the sensor, which is similar to the process presented in [18]. The sensor was fabricated using a silicon on insulator (SOI) wafer. Both the device layer and the handle layer of the SOI wafer were highly doped, with resistivity less than 0.005 Ω cm. The process started by wet thermal oxidation at 1150 °C to grow a 2 μm thick layer of silicon oxide on both sides of the wafer, see Figure 5a. The pattern for the electrodes was then transferred into the top oxide layer. Next, deep reactive ion etching (DRIE) was used to transfer the pattern into the device layer, see Figure 5b. The same process of patterning the oxide followed by DRIE was then repeated for the backside of the wafer. The DRIE process at the back side of the wafer etched the inlet and outlet holes as well as a trench around the chip, see Figure 5c,d. Then, both the thermal oxide layers and the BOX layer were etched, first 2 min in 50% HF solution and then 45 min in vapor HF to prevent stiction, see Figure 5e. During the vapor HF step the chips were released from the wafer as described in [18,19,20]. In parallel, glass caps were made by etching fluidic channels in 50% HF and then diced to fit the required size. The glass caps were glued onto the silicon chips using PDMS, see Figure 5f. PDMS glue also fills the trenches that lead to the bond pads of the chip. Figure 5g shows an SEM image of two sensors at a fluid inlet/outlet.

## 4. Measurement Method

Figure 6 shows a schematic drawing of the measurement setup and a photo of the chip mounted on a PCB. An HP 4194a impedance analyzer was connected as indicated to measure the impedance between two sensor electrodes. The chip was connected to a PCB by wire bonding and the PCB was connected to the impedance analyzer by four 20 cm coaxial cables with MMCX connectors. The handle layer of the chip was grounded by wire bonding.

Nitrogen, ethyl acetate, 1-hexanol, isopropanol, ethanol and ethylene glycol were used in the measurement to cover a relative permittivity range from 1 to 42. No aqueous solutions were used in this work because aqueous solutions have a high ion density and form an electrical double layer at electrode surfaces. The complex behavior of aqueous solutions is out of the scope of the current research. The liquids were pushed into the measurement system using a syringe pump. The syringe pump first pumps enough liquid into the sensor after which the pump was stopped and the measurements were performed. Nitrogen was introduced into the system by a compressed gas source. The nitrogen flow was also stopped when the measurements were performed. The device was flushed with isopropanol and DI water and purged with nitrogen between each measurement to remove residual chemicals. The amplitude of the measurement voltage $V_{i}$ was set to be 0.5 V. A logarithmic frequency sweep from 100 kHz to 1 MHz was performed to measure the impedance of the sensor. All measurements were performed at room temperature and repeated 10 times for each fluid.

## 5. Measurement Results and Discussion

Figure 7 shows the measurement results in the form of a Nyquist plot. Both the real part $Z^{\prime }$ and imaginary part $Z^{\prime \prime }$ of the impedance decrease with increasing frequency, as expected from Equations (Equation 2), (Equation 3) and (Equation 4). We can rewrite these equations into
(6)$${Z^{\prime \prime }}^{2}+{\left(Z^{\prime }-\frac{R_{1}}{2}\right)}^{2}={\left(\frac{R_{1}}{2}\right)}^{2}$$

Hence, each curve in Figure 7 should be part of a circle with the center at $Z^{\prime }=R_{1}/2$ and $Z^{\prime \prime }=0$ if both the x-axis and y-axis have the same scale. In our measurement $Z^{\prime \prime }$ is much larger than $Z^{\prime }$ for all frequencies, indicating that $R_{1}$ is much larger than $X_{C_{1}+C_{P}}$ and the impedance is mostly determined by $C_{1}$ and $C_{P}$. Fitting Equation (Equation 6) to the measurement data shows that $R_{1}$ depends on the fluid with values between 3.4 and 100 MΩ.

This is confirmed by Figure 8, which shows the absolute value of impedance as a function of frequency. A clear capacitive behavior is observed with the impedance inversely proportional to frequency. Figure 7 and Figure 8 both contain the results of all 10 frequency sweeps per fluid, however the traces can hardly be distinguished, indicating good reproducibility of the measurements.

Figure 8 was used to extract the capacitance value for each of the fluids. The values are listed in Table 3 together with the simulated value based on the model shown in Figure 4 and the relative permittivity found in [21]. Figure 9a shows the measured and simulated capacitance (red line) versus relative permittivity. The measured result shows great linearity with the reference relative permittivity. Figure 9b shows the linear fit residual error of Figure 9a. The residual error for each fluid is smaller than 0.9% of the full scale. The standard deviation for each fluid is smaller than 0.05% of the full scale. However, the simulation results as shown by the red line deviate from the measurement results. Two reasons may cause this error. Firstly, the handle layer is not a perfect conductor. Thus some capacitance through the BOX layer and handle layer influences the measurement. Secondly, due to surface tension, the electrical insulation trenches around the anchor point might be partially filled with PDMS when the top glass cap was glued on the chip. The blue line in Figure 9a shows the simulated capacitance when there is a 15 μm thick layer of PDMS at the bottom of the electrical insulation trench. The slope of the blue line is close to the slope of the linear fit of the measurement results. By extrapolating the linear fit to a relative permittivity of zero, we find the value of parasitic capacitance $C_{P}$ = 0.735
pF.

## 6. Conclusions

A micro relative permittivity sensor that allows fluid to flow through the sensing element has been designed, simulated and fabricated. In tests with different fluids with relative permittivity ranging from 1 to 42, the sensor shows excellent linearity with maximum fit error of 0.9% of the full scale. In comparison with conventional designs, the sensor is very small but still provides a high capacitance value that allows electrical measurement to be able to determine small changes in the relative permittivity of the fluid. Future research will focus on measurements with aqueous solutions, design of new sensors that allow on-chip four point capacitance measurement and multiparameter sensor integration.

## Figures and Tables

**Figure 1 micromachines-11-00325-f001:**
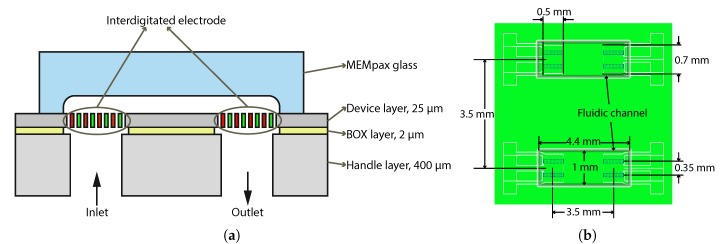
(**a**) Schematic cross-sectional drawing of the sensor. Fluidic inlets and outlets are realized through the handle layer of an SOI wafer. Interdigitated electrode pairs are suspended in the device layer above the inlet and outlet. A fluidic channel between inlet and outlet is realized by means of a MEMpax glass cap. (**b**) Schematic top view of the sensor (not to scale). The chips contains a total of eight electrode pairs that can be connected independently from each other.

**Figure 2 micromachines-11-00325-f002:**
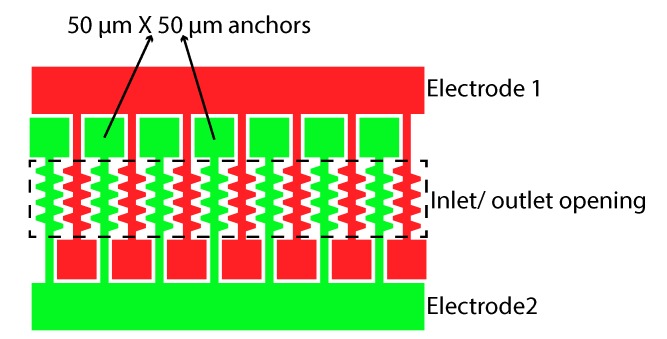
Design of the interdigitated electrode pattern. The dashed line indicates the opening in the handle layer. The electrodes are attached to the handle layer at both sides of the opening by means of 50 μm × 50 μm anchors. As a result the heart-to-heart spacing between the electrodes is 35 μm and fingers are attached to the sides of the electrodes to further increase the capacitance. The total length of the electrode is 210 μm. The width of the electrode is 10 μm without trapezoid shape fingers. The height of the electrode is similar to the thickness of the device layer, which is 25 μm.

**Figure 3 micromachines-11-00325-f003:**
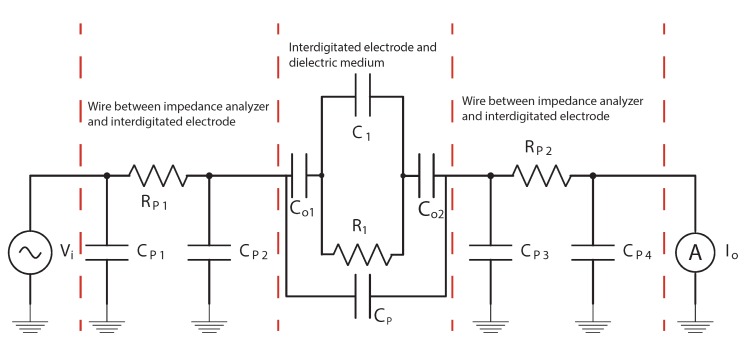
Equivalent circuit of the sensor including parasitic capacitance and series resistance.

**Figure 4 micromachines-11-00325-f004:**
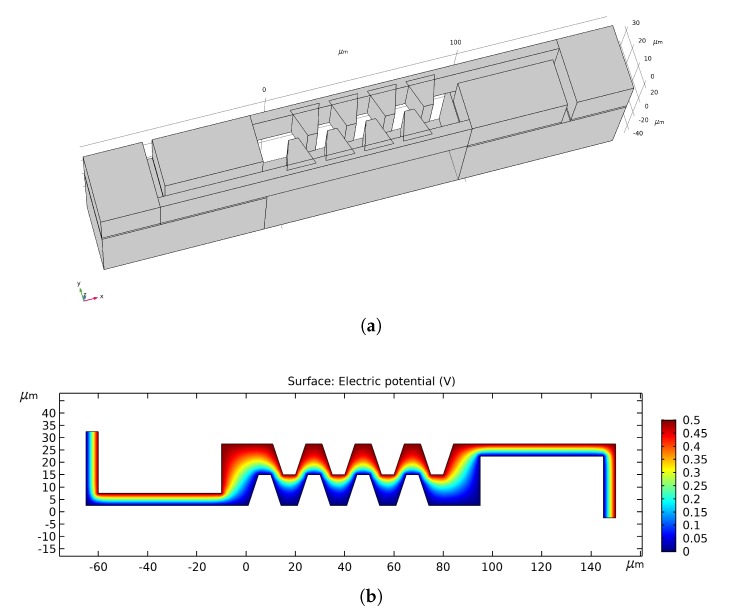
(**a**) Electrode geometry used for finite element simulations. The geometry is repeated 13 times above each fluid inlet and outlet as shown in Figure 2. (**b**) Simulated electric potential 12.5 μm below the chip surface. The voltage between the two electrodes is set to 0.5 V. The liquid between the electrodes is set to isopropanol.

**Figure 5 micromachines-11-00325-f005:**
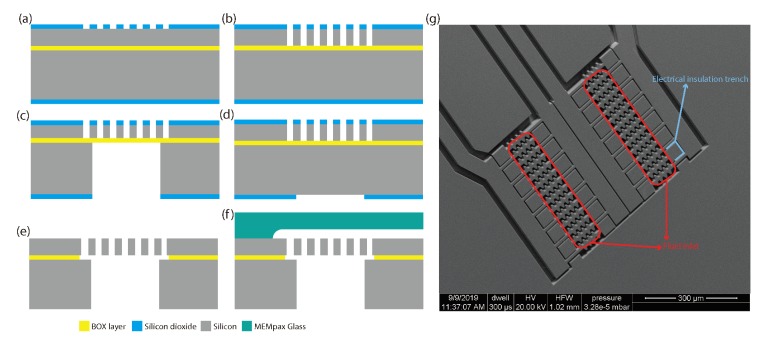
(**a**–**f**)Fabrication process of the relative permittivity sensor. (**g**) SEM image of a fluid inlet/outlet and the interdigitated electrode pairs above the two fluid openings. The silicon connectors going to the bond pad are isolated by etched trenches. These trenches are filled with PDMS when the cap is glued on.

**Figure 6 micromachines-11-00325-f006:**
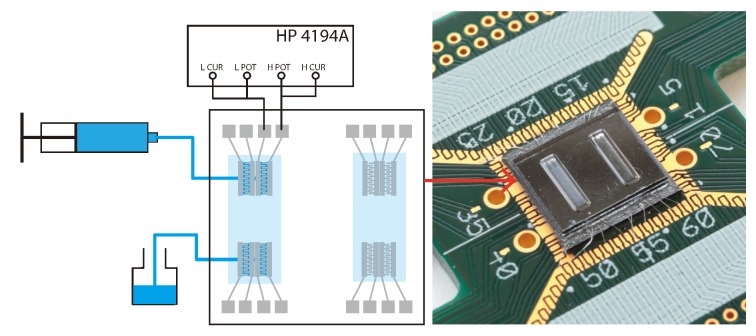
Schematic drawing of the measurement setup. The blue lines represent the fluid path. The light blue areas represent the fluid channel on chip. The photograph shows the chip glued and wire bonded on a printed circuit board.

**Figure 7 micromachines-11-00325-f007:**
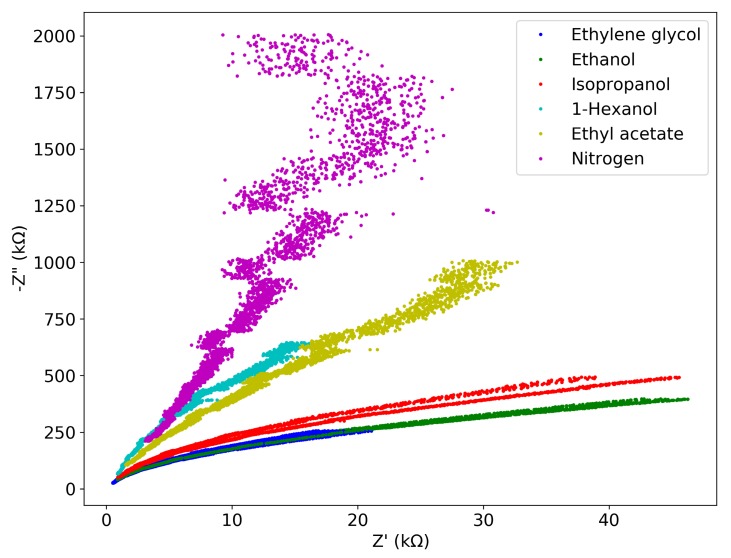
Nyquist plot for each selected fluids between 100 kHz and 1 MHz. On each curve, $-Z^{\prime \prime }$ decreases as the frequency increases. Note the difference in scale between x-axis and y-axis.

**Figure 8 micromachines-11-00325-f008:**
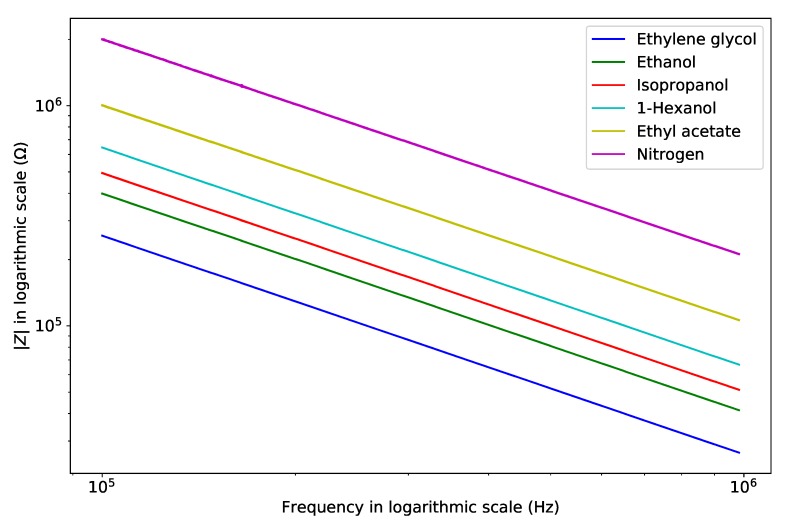
Measured absolute value of impedance as a function of frequency. The results clearly show a capacitive behavior with impedance inversely proportional to frequency.

**Figure 9 micromachines-11-00325-f009:**
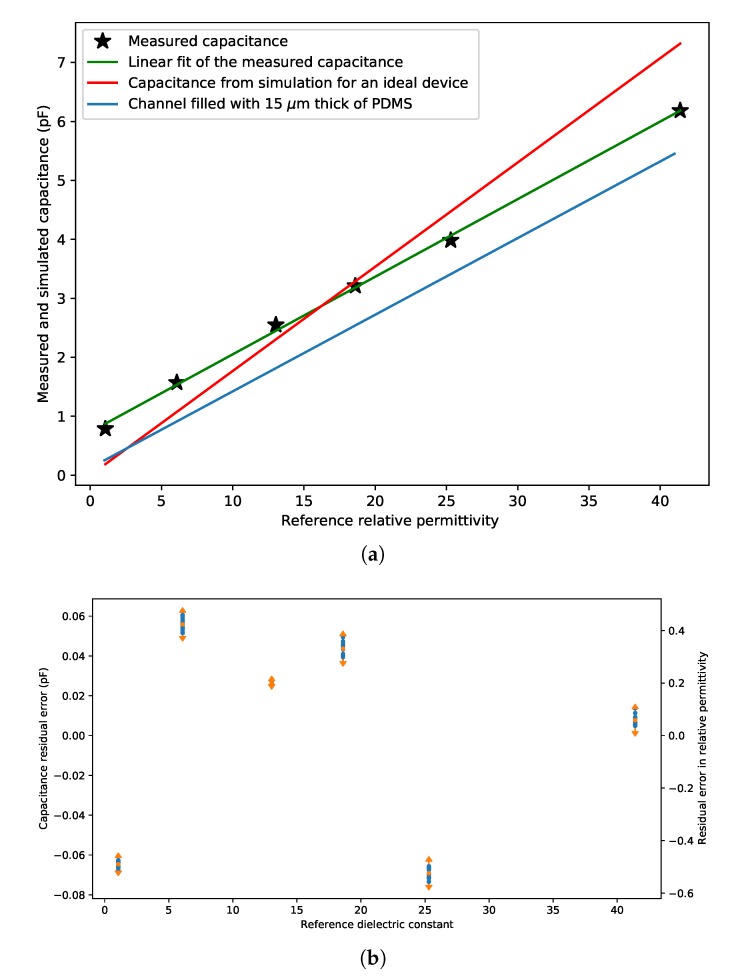
(**a**) Measurement and simulation capacitance versus reference relative permittivity. Black stars represent the capacitance measured with different fluids. The red line represents the simulated result without the correction for PDMS inside the electrical insulation channels. The blue line represents the simulated result when the electrical insulating channels are partially filled with PDMS. (**b**) Linear fit residual error of (**a**). The length of the error bars equal four times the standard deviation.

**Table 1 micromachines-11-00325-t001:** A comparison between published sensors and the sensor as presented in this paper.

Device	Sensor Dimensions (mm × mm)	Distance between Electrodes (μm)	Measured Capacitance Range (pF)	Measured Relative Permittivity Range
Demori et al. [2]	0.3 × 10	300	0.152–0.164	1–80
Shih et al. [1]	0.2 × 0.2 or 0.1 × 2	4	0.3–1.2	18.6–80
Aslam et al. [13]	12 × 100	4000–12,000	10–35	2–25.4
Isgor et al. [15]	0.45 × 0.8	50	0.05–0.275	2.5–80
Lötters et al. [3]	0.2 × 1	roughly 40	0.7–1.8	1–80
Chretiennot et al. [6]	0.18 × 1.3	20	Not a capacitance sensor	Not applicable *
Alveringh et al. [16]	1 × 1	40	0.01–0.16	1–32.7
This paper	0.5 × 0.7	12.5	0.8–6.2	1–41.4

* This device measures impedance at 20 GHz.

**Table 2 micromachines-11-00325-t002:** Measured or calculated component values for the equivalent circuit in Figure 3.

Component(s)	Sensor Capacitance (pF)
$C_{P1}$, $C_{P4}$ Parasitic capacitance of the PCBs and cables	38.7 pF (measured)
$C_{P2}$, $C_{P3}$ Parasitic capacitance of conductive tracks and bond pads	3.1 pF (calculated)
$R_{P1}$, $R_{P2}$ Series resistance of the conductive tracks and bond wires	<48 Ω (calculated)
$C_{P}$ Parasitic capacitance between the electrodes	0.7 pF (measured)
$C_{1}$ Capacitance of the fluid medium	0.2–6 pF (measured, depending on fluid)
$R_{1}$ Resistance of fluid	3.42 MΩ–0.1GΩ (measured, depending on fluid)
$C_{o1}$ and $C_{o2}$ Capacitance of native oxide layer on electrodes	roughly 0.32 μF (calculated)

**Table 3 micromachines-11-00325-t003:** The measured sensor capacitance, simulated sensor capacitance and reference relative permittivity of different chemical substance.

Chemical Substance	Sensor Capacitance (pF)	Simulated Capacitance (pF)	Reference Relative Permittivity
Nitrogen	0.788	0.187	1.06
Ethyl acetate	1.571	1.07	6.08
1-hexanol	2.549	2.3	13.03
Isopropanol	3.211	3.29	18.6
Ethanol	3.983	4.47	25.3
Ethylene glycol	6.184	7.32	41.4

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
