# Peer review of "A Flow-Through Microfluidic Relative Permittivity Sensor"

_micromachines, 2020, doi:10.3390/mi11030325_

Round 1

Reviewer 1 Report

Line 11: The author uses the exact words in the title of the reference 2, “fluid discrimination and characterization”. Paraphrasing is required.

Line 18: Either the line starting with “These Designs are small in size” needs to be a separate sentence on itself or the entire sentence needs to be punctuated/rearranged.

Line 18: Please be more specific when describing the dimensions. Use of adjectives like “small” needs to be avoided and more specific dimensions/geometry need to be provided.

Line 19: Same as the comment above, “much larger capacitance” and “less parasitic” can be avoided or compared to an existing design to indicate how much larger  or less than “a particular comparative parameter”

Line 19: Use of “and” at the end of the line is not necessary.

Line 22: “Furthermore” is a single word.

Line 22: Reference supporting the statement needs to be provided.

Line 27: Sentence needs to be rewritten with specifics of how the device can be compatible in a sensor fusion setup. The word “combination” is not appropriate for the claim made by the author.

Line 33: The line needs to be rewritten. The sentence describing two parallel openings “both” for inlet and outlet is confusing in reference to the figure. If it is just two trenches etched in parallel to form the inlet and outlet, the word “both” can be removed.

Line 36: The word “a” can be removed before the word “red”.

Line 37: Change of grammatical tense is not recommended. Please stick to one “tense” for all the device description writing.

Figure 1: The figure description uses the exact same words as the main text body. Figure description can be changed to be more precise, since the device is described in detail in the previous few sentences above the figure.

Line 42: This sentence is really not clear. Author needs to be more detailed in describing the device layout. Stating that “the fingers are added to the sides of the electrodes” doesn’t make any sense.

Figure 2: The description needs to be more clear and concise. Also, the dimensions of the interdigitated fingers needs to be mentioned apart from the center-to-center distance. The width, length and height parameters can provide a more detailed overview of the proposed device.

Line 44-51: All the details provided in this paragraph can be consolidated and provided as an additional column in Table 1.

Line 57: The results of the COMSOL simulations could be added to Figure 4.

Line 60. Use of full stop after 0.187 pF.

Line 64: Please provide the doping concentration and conductivity measurements.

Line 68: Sentence can be rewritten.

Line 67: Providing the oxide thickness and etch profile would provide the reader a detailed understanding of the fabrication process.

Line 90: “Fluid” and not plural.

Line 100: Can the measurements be repeated ensuring proper grounded handle?

Figure 9: The Authors claim that measurements were repeated 10 times. Could you provide the standard deviation and error bars for each fluid individually? Does the trend of the measured capacitance vs relative permittivity change with each repetition?  

Line 108: The claim that change in trend is due to the deposition of PDMS on the electrical insulation needs to be supported with more experiments. For example, since it is a glass surface attached to the Si wafer, an oxygen plasma activation or any other dry bonding technique can be used to create stiction between the glass cover and Silicon. This way, if the slope of measured capacitance agrees with the simulation in the absence of PDMS, then the claim that the PDMS causes this change in trend can be validated.

Line 177: Please be specific when comparing the fabricated device with commercial or existing devices. The use of words like “very small” in conclusions must be avoided. Also, a high capacitance value in relation to what? Even “fF” capacitance can be considered a high capacitance in certain geometries and applications. A general use of term “high capacitance” is not appropriate. Again, claiming that “electrical measurements to be performed easily” is very vague and needs to be properly rewritten.

Author Response

Thank you so much for your detailed comments.

Reviewer 2 Report

This paper reports on the extraction and verification of relative permittivity to the six different samples, e.g., ethylene glycol, ethanol, isopropanol, 1-hexanol, ethyl acetate, nitrogen, using a microfluidic channel system based on chip level. Authors have conducted all procedures, i.e., design, simulation, fabrication, and characterization, for research. I think that these are the results collaborating with each other well. So this paper is very well organized and written.

However, I would like to indicate some questions and minor comments.

First, in Table 2, the authors represented the simulated capacitance. I wonder these values were just obtained from either the electromagnetic solver, i.e., COMSOL multi-physics simulation or the circuit simulation based on the circuit model, as shown in Fig. 3.

Second, the authors used the syringe pump to push into the measurement system. Represent the flow ratio to the liquid sample of syringe pumps (or condition of syringe pump) for liquid sample injection.

Minor comments:

The authors must remove the double-used preposition as follows. In line number 39, “for” was used doubly. And in line number 93, “in” was also used doubly.

Author Response

Thank you very much for your comments.

Round 2

Reviewer 1 Report

Thank you for addressing all the suggested comments! 

Best

Dr. Rajan